# A New Generation of OPM for High Dynamic and Large Bandwidth MEG: The ^4^He OPMs—First Applications in Healthy Volunteers

**DOI:** 10.3390/s23052801

**Published:** 2023-03-03

**Authors:** Tjerk P. Gutteling, Mathilde Bonnefond, Tommy Clausner, Sébastien Daligault, Rudy Romain, Sergey Mitryukovskiy, William Fourcault, Vincent Josselin, Matthieu Le Prado, Agustin Palacios-Laloy, Etienne Labyt, Julien Jung, Denis Schwartz

**Affiliations:** 1CERMEP-Imagerie du Vivant, MEG Departement, 69000 Lyon, France; 2CRNL, UMR_S 1028, HCL, Université Lyon 1, 69000 Lyon, France; 3MAG^4^Health, 38000 Grenoble, France; 4CEA LETI, Minatec Campus, Université Grenoble Alpes, 38000 Grenoble, France

**Keywords:** OPM, MEG, SQUID, brain activity, atomic magnetometer, helium OPM, neuroimaging

## Abstract

MagnetoEncephaloGraphy (MEG) provides a measure of electrical activity in the brain at a millisecond time scale. From these signals, one can non-invasively derive the dynamics of brain activity. Conventional MEG systems (SQUID-MEG) use very low temperatures to achieve the necessary sensitivity. This leads to severe experimental and economical limitations. A new generation of MEG sensors is emerging: the optically pumped magnetometers (OPM). In OPM, an atomic gas enclosed in a glass cell is traversed by a laser beam whose modulation depends on the local magnetic field. MAG^4^Health is developing OPMs using Helium gas (^4^He-OPM). They operate at room temperature with a large dynamic range and a large frequency bandwidth and output natively a 3D vectorial measure of the magnetic field. In this study, five ^4^He-OPMs were compared to a classical SQUID-MEG system in a group of 18 volunteers to evaluate their experimental performances. Considering that the ^4^He-OPMs operate at real room temperature and can be placed directly on the head, our assumption was that ^4^He-OPMs would provide a reliable recording of physiological magnetic brain activity. Indeed, the results showed that the ^4^He-OPMs showed very similar results to the classical SQUID-MEG system by taking advantage of a shorter distance to the brain, despite having a lower sensitivity.

## 1. Introduction

MagnetoEncephaloGraphy (MEG) is a non-invasive functional imaging technique providing a direct measure of neuronal activity at a millisecond time scale. From these signals, one can derive the dynamics of normal or pathological brain networks [1]. Conventional MEG systems (SQUID-MEG) use superconducting quantum interference devices (SQUIDs) to reach adequate sensitivity to map brain activity [2]. However, SQUIDs require a very low temperature (4.2 K) which is achieved using liquid helium. Beyond a very high cost and a strong environmental impact, it leads to severe experimental limitations: The sensors are enclosed within a rigid dewar with a vacuum space separating sensors from the scalp. Thus, the helmet is not adaptable to the geometry of a given head. The sensor-to-brain distance is around 3 cm for adults and increases for smaller heads, inducing a substantial drop in the intensity of the recorded signal. The subject’s head is not fixed with respect to the sensors and any movement can reduce data quality markedly. The SQUID-MEG systems are bulky, making the experimental environment unnatural and severely limiting the experimental paradigms that can be utilized. These limitations impede the use of MEG for many subject groups, including children and patients with various pathologies.

A new generation of MEG sensors is emerging: the optically pumped magnetometers or OPM. In OPM, an atomic gas enclosed in a glass cell is traversed by a laser beam whose modulation depends on the local magnetic field.

Alkali-based OPMs were developed first [3]. They have very good sensitivity (15 fT/√Hz in dual-axis mode) and can be placed near the scalp, potentially allowing a three-to-eight-fold increase in signal power of neuromagnetic activity recording [4,5,6,7]. These sensors have been successfully used in a wide range of experimental studies involving children and adults, both in healthy volunteers [8,9,10,11] and patients [12,13]. A complete review of the field can be found in [14,15]. Note that these studies also showed that these OPM sensors could be easily used in a far wider field of applications than classical SQUID sensors, for example, recording activity from the spine or the retina [16,17]. Compared to classical MEG, these sensors have the advantage of outputting multi-axis measurements of the brain magnetic field, potentially giving access to previously non-recorded brain activities thanks to a better description of high spatial frequency structure in the magnetic field [7,8,18]. Alkali being in a solid state at room temperature, requires heating to 150 °C to generate an alkali vapor and achieve its operating mode. Consequently, they dissipate 0.7 W per sensor. They have a limited bandwidth (1–100 Hz) and a small dynamic range (5 nT). This necessitates the use of optimized shielded rooms reducing the remnant field [11,19,20]. It may impose the use of an additional system of field nulling coils around the patient’s head to compensate for the remaining magnetic field inside the magnetically shielded room to reduce the cross axes projection error and inhomogeneity between sensors within the array [7,19,21,22,23] depending on the level of environmental noise and subject movement.

Recently, a promising alternative to alkali OPMs has emerged: MAG^4^Health is developing OPMs using Helium gas (^4^He-OPM) [24]. ^4^He-OPMs were successfully applied to the recording of brain magnetic fields [25]. The ^4^He-OPMs output, natively, a 3D vectorial measure of the magnetic field. They have key advantages: (i) ^4^He-OPMs operate at room temperature without heating (Helium is a gas at room temperature), so there is no noticeable heat dissipation (0.02 W per sensor); (ii) ^4^He-OPMs have a large resonance linewidth which in combination with a closed loop control gives access to a large dynamic range allowing for lightweight shielding and subject’s movement (>200 nT) and a large frequency bandwidth adapted to the brain electrical activity (0–2000 Hz). The closed loop control continuously cancels the three components of the magnetic field of each sensor by applying an opposite compensation field along three axes. This specific operating mode guarantees a very stable and homogenous sensor accuracy within the array and contributes to the large dynamic range.

The main drawback of the very first version of the ^4^He-OPMs [25] used for the proof of concept of MEG recording was their lower sensitivity (200 fT/√Hz). However, MAG^4^Health has made considerable progress in this aspect. Recent results show that the ^4^He-OPMs now reach a sensitivity of better than 45 fT/√Hz on two of the three axes, with a very limited 1/f noise rise [26]. Currently, the sensor is packaged in a 3D printed mount made from a photosensitive resin, with a footprint of 2 × 2 × 5 cm weighing 45 g without the cabling. In this study, the cables were supported by a wooden frame fixated on the subject’s chair.

In this study, five ^4^He-OPMs were tested together. Our main aim was to compare these new ^4^He-OPMs sensors to our classical SQUID-MEG system to evaluate their experimental performance in a larger sample than is commonly used in similar studies, consisting of 18 healthy volunteers. Considering that the ^4^He-OPMs operate at real room temperature (no heating, no cooling) and can be placed directly on the subject head, our assumption was that the current version of ^4^He-OPMs would provide a reliable recording of physiological magnetic brain activity. Indeed, the results show that the ^4^He-OPMs, despite having a lower sensitivity and reduced noise reduction potential due to the small number of sensors, yielded very similar results to the classical SQUID-MEG system by taking advantage of a shorter distance to the brain.

## 2. Materials and Methods

### 2.1. Data Acquisition

All MEG data were collected in a standard magnetically shielded room (MSR) (2 µ-metal layers, 1 copper layer, Vacuumschmelze, Hanau, Germany) without any active shielding. The background magnetic field of this chamber is approximately 20 nT.

### 2.2. Classical SQUID-MEG

Recordings were performed with a 275 SQUID-based axial gradiometers MEG system (CTF MEG Neuro Innovations Inc., Port Coquitlam, BC, Canada). The subjects were comfortably seated with the head immobilized by an inflatable cushion. The MEG signal was digitized at 2 kHz with a 600 Hz low pass filter for the somatosensory experiment and at 1 kHz with a 300 Hz low pass filter for the visual experiment.

### 2.3. ^4^He-OPMs

Recordings were performed using newly developed ^4^He-OPM sensors measuring the brain’s magnetic field along 3 axes (1 radial to the scalp and 2 tangential) with continuous self-compensation [26]. The brain magnetic field measurement relies on a measure of the light intensity modulation caused by the deviation of the electronic spin of ^4^He atoms from their alignment imposed by laser pumping. The ^4^He-OPMs array used in this work consists of 5 optically pumped magnetometers based on the parametric resonances of ^4^Helium metastable atoms in a near zero magnetic field [27,28,29]. The size of the cell containing the ^4^He gas atoms is cylindrical with a 1 cm internal diameter and 1 cm internal height. The bottom of the sensor is surrounded by small 3-axis Helmholtz coils, which are used to apply both the RF fields and the compensation fields. A dynamic range of ±250 nT is currently achieved by ^4^He-OPMs. The sensitivity of magnetometers operating in the closed-loop tri-axial mode is better than 45 fT/√Hz on two of the three axes (one radial and one tangential) with a bandwidth going from DC to 2 kHz. The third axe (radial) has a sensitivity 4 times lower (200 fT/√Hz). A more detailed description of the sensor can be found in Appendix B and a previous publication [26]. Because sensitivity is optimal on two of the three axes, all analyses were done on these axes.

The subject was comfortably seated, wearing a conformable headset housing 96 possible positions for the ^4^He-OPM sensors (see Figure 1). Head movements were possible, although the participant was asked to avoid large head movements. One ^4^He-OPM sensor was located 10 cm above the head of the subject to serve as a reference sensor measuring the ambient noise in the MSR and the magnetic artifacts generated by the subject movements. The reference was fixed on top of a pillar which was positioned on the Cz slot of the headset. For the somatosensory experiment, the 4 remaining ^4^He-OPM sensors were located in LC11, LC13, LC 31 and LC33 locations around the somatosensory area (see Figure 1). For the visual experiment, the 4 remaining ^4^He-OPM sensors were located in LO11, LO31, RO11 and RO31 locations around the primary visual area (see Figure 1). The signal was sampled at 11 kHz. No subjects reported any discomfort related to the weight of the system. One-third of the subjects mentioned tension/traction related to the cables and their fixation on the wooden support restricting their freedom of movement.

### 2.4. Participants and Experimental Design

This study was approved by regulatory and ethics administrations in France (IDRCB n° 2020-A01830-39). Subjects signed a written informed consent prior to participation. The study included 18 healthy subjects taking part in two tasks: a somatosensory stimulation session (17 subjects) and a visual stimulation session (all 18 subjects). All participants were aged 18–60 (mean 34.2, std 8.2) years. They had no history of neurological or psychiatric disorders and were not taking any medications active in the central nervous system.

Somatosensory experiment: The median nerve was electrostimulated transcutaneously on the right wrist by using a bipolar electrode connected directly to the stimulator (S88 stimulator, Astro-Med Inc. GRASS, W. Warwick, RI, USA). The motor threshold (MT, the minimal stimulus intensity required to produce thumb movement) was determined for each subject, and the experimental stimulus intensity was set at MT + 10% [30]. Eight hundred stimulations (0.5 ms shocks) were delivered, with a randomized inter-stimulus interval varying from 350 to 450 ms. The subjects were asked to keep their eyes open while watching a silent movie.

Visual experiment: After a baseline with central fixation (1.2 s ± 0.2 s), a Gabor visual stimulus (4 degrees of visual angle in diameter, spatial frequency of 12 cycles across the stimulus at full contrast, 3 cycles/deg) was presented centrally to the subject using a Propixx video projector (VPixx, Saint-Bruno, QC, Canada) at a distance of 82 cm. Using a 2-button response pad, the subjects were instructed to discriminate orientation difference direction of the Gabor relative to a reference orientation, which could be either large (25 deg, considered ‘easy’) or small (15 deg, considered ‘difficult’). In total, there were, therefore, 4 types of stimuli, although this distinction was not made in the current analysis. The stimulus remained on the screen until a response was given or maximally 2.5 s. In total, the experiment consisted of 900 trials. The orientation task insured a constant and reliable level of attention. It is to be noted that the transition probability between the 4 types of stimuli was further manipulated, but the effect of such manipulation was not analyzed here.

### 2.5. Data Analysis

All data were analyzed using MNE-python (version 1.1.1 [31]) in a Linux environment and were as similar as possible for SQUID-MEG and ^4^He-OPM data while optimizing data quality for each modality. For both datasets, breaks during acquisition longer than 9 s, as well as spiking artifacts (fast amplitude deviations) larger than 3 standard deviations (sample-to-sample), were marked to be ignored. Due to the presence of low frequency drifts in the ^4^He-OPM data, (slow) amplitude deviations larger than 5 standard deviations (relative to signal average) were also removed. For SQUID-MEG data, third-order gradient compensation was applied [32]. For ICA decomposition only, the data were downsampled to 500 Hz and bandpass filtered between 1–100 Hz. ICA components related to eye movements, blinks, heartbeat and gross non-neural artifacts were removed. For SQUID-MEG, on average, 4.8 (4.5/5.2 somatosensory/visual) components were removed, and 2.8 (both tasks) components for ^4^He-OPMs. The increased amount of rejected components for SQUID-MEG is due to the increased number of sensors in the data, allowing for better typification of eye- and heart-related artifacts. For the somatosensory task, the stimulation artifact was removed by mean-interpolating 10 ms before and after stimulation (t = 0 s). Data were subsequently band-pass filtered between 1 and 100 Hz, with an additional line noise filter at 50 and 100 Hz. Electro-oculogram (EOG) recordings were acquired for the visual task, which was used as a control regressor for residual eye-movement-related artifacts. Additionally, for the ^4^He-OPM data in both tasks, the radial and one tangential axis data from the reference sensor was used as a control regressor for non-neural environmental noise.

Evoked field: Data were epoched with 100 ms baseline and 250 ms post-stimulus for the somatosensory task and 200 ms baseline and 500 ms post-stimulus for the visual task. For SQUID-MEG data, further analyses were restricted to the 4 sensors closest to the positions of the ^4^He-OPM montage. To find these closest sensors, the 96 possible locations of the OPM headset were digitized in the SQUID-MEG subject reference frame (based on three fiducial markers: nasion, left and right pre-auricular points) using a Polhemus system (Colchester, USA) in 1 subject serving as a template. For the somatosensory experiment, these were MLC25, MLF64, MZC02 and MLP11 and for the visual experiment, MRO31, MRO11, MLO31 and MLO11 of the CTF sensor layout (see Figure 1). Epochs containing gross artifacts were rejected using the autoreject package (version 0.4 [33]) for MNE. The total rejected portion of trials was forced to be 70% or lower to ensure sufficient trials for quality estimation. For the somatosensory task, 93.8% or 750 trials of the SQUID-MEG and 88.2% or 705 trials of the ^4^He-OPMs data were retained on average. For the visual task, 96.0% (921 trials) of the SQUID-MEG data and 81.7% (784 trials) of the ^4^He-OPM data were retained on average. Note that for the ^4^He-OPM data, rejection was based on the 2 best-performing axes (radial and one tangential).

Oscillatory power: To test the performance of the sensors in oscillatory brain dynamics, the visual task was chosen due to its long time window and established expected oscillatory pattern, being a stimulus-induced decrease in alpha-band (8–13 Hz) oscillations and increased gamma band (~30–80 Hz) oscillations [34]. Therefore, for the visual task only, a slightly longer time window was chosen (relative to the evoked field analysis) to calculate the oscillatory power [−0.6 s, 0.6 s] to avoid edge artifacts. The oscillatory power in the 2–100 Hz range was then calculated using Morlet wavelets with a fixed length of 250 ms. Statistical significance post-stimulus onset was calculated using two-tailed cluster-based permutation [35] testing between the baseline interval [−0.4 s, 0 s] and the post-stimulus time window [0 s, 0.4 s] with 1000 permutations and an initial cluster threshold of *p* = 0.05.

Signal-to-noise ratio (SNR) was calculated as the ratio between the maximum absolute value of the evoked response in the post-stimulus interval ([0.02 s, 0.25 s] for somatosensory stimulation, [0 s, 0.3 s] for visual stimulation) and the average baseline ([−0.1 s, −0.02 s] for somatosensory, [−0.2 s, 0 s] for visual) standard error over epochs. Time windows were chosen to be as long as possible while avoiding artifactual contamination. Note that there was no delayed response in the visual task, and responses started to occur > 300 ms. SNR values were computed for each sensor and subject individually. To ensure a comparable estimation, the SNR estimate was corrected for the number of trials in the individual dataset by taking a subset of trials from the modality containing a larger number of trials. This was repeated (1000 iterations) to eliminate selection bias.

Empty room recordings: To quantify the raw performances of SQUID-MEG and ^4^He-OPMs sensors, we performed empty room recordings with identical acquisition parameters as described above except for the recording duration, which was set to 30 s. These SQUID-MEG and ^4^He-OPMs signals were then band-pass filtered between 1 Hz and 300 Hz with a notch filter at 50 Hz. For each sensor, a PSD was computed using Welch’s method. We then computed for SQUID-MEG and ^4^He-OPMs a mean sensitivity value by averaging PSD values across sensors and across a frequency band ranging from 5 Hz to 90 Hz. Note that for comparison purposes, baseline PSDs were also computed on the 500 ms preceding the stimulation for the visual task.

## 3. Results

### 3.1. Empty Room Results

The empty room recording for ^4^He-OPMs showed a mean PSD value of 42.65 ± 2.97 fT/√Hz and a mean PSD value of 3.36 ± 1.08 fT/√Hz for the SQUID-MEG. Figure 2 shows that both kinds of sensors retain their respective sensitivity across the frequency band (up to 300 Hz). Similar peaks due to the power line (and its harmonics) and due to other environmental causes appear at the same frequencies on both PSDs except for an additional peak around 130 Hz on the ^4^He-OPMs PSD.

### 3.2. Somatosensory Stimulation

Event-related fields. Figure 3 shows the group-averaged responses to the median nerve stimulation from sensors located above the somatosensory cortex for both SQUID-MEG and ^4^He-OPMs, revealing a very similar time course for both measurements. The Pearson product-moment correlations at lag zero between the RMS time course for SQUID-MEG and ^4^He-OPMs were r = 0.91 for the radial component and r = 0.92 for the tangential component, confirming a very high degree of similarity. In response to the somatosensory stimulation, we observed somatosensory evoked fields with deflections at 20, 35, 80 and 135 ms (21/36/79/135 ms for SQUID-MEG, 20/34/84/136 ms for radial ^4^He-OPMs and 20/36/80/135 ms for tangential ^4^He-OPMs), which is consistent with N20, P35 and N130 deflections from ERP literature [36,37]. The average maximum deflection values for ^4^He-OPMs reached 328 fT, whereas SQUID-MEG reached 71 fT, corresponding to a more than 4.5-fold increase. This is consistent with our expectations regarding the ^4^He-OPM sensor, given its proximity to the scalp.

Figure 4 directly compares the event-related fields for the SQUID and ^4^He-OPMs sensors. For each subject, SQUID and ^4^He-OPMs sensors (in both radial and tangential directions) were chosen with the highest signal-to-noise ratio. The top three panels show individual results with varying degrees of correspondence to the SQUID-MEG signal. As one can see, there is a high level of correspondence between sensor types, especially when noise levels are low, suggesting that nearly identical results can be obtained under ideal circumstances. This is supported by high degrees of correlation between time courses (up to r = 0.95), although individual variance can be substantial (see Appendix A Figure A1 for an overview of all subjects).

Signal-to-noise ratio. To assess the data quality obtained from the SQUID-MEG and OPM-MEG measurements, the signal-to-noise ratio (SNR) was calculated per modality and sensor type. As depicted in Figure 5, the SNR values show a clear overlap between SQUID-MEG and ^4^He-OPMs, especially for the radial axis (SNR 18.2 SD 8.2). SQUID-MEG benefits from the highest SNR (26.5 SD 11.0) but shows the largest range between subjects. This difference is significant (paired samples *t*-test: *t* = 3.72, *p* < 0.005).

### 3.3. Visual Stimulation Experiment

Event-related fields. For the visual stimulation task, the group averaged time courses are depicted in Figure 6. The onset of the visual stimulus was at t = 0. Sensors were located above the left and right visual areas. In response to the onset of the Gabor grating, a clear event-related field, peaking around 100 ms, corresponding to the expected P100 [38], can be reliably observed at all sensors. The average latency for SQUID-MEG is 89 ms, whereas, for the ^4^He-OPMs, latency is at 85 and 77 ms for the radial and tangential components, respectively. This difference in average latency is not significant. Squid-MEG maximum amplitude for the P100 reaches 65 fT, while the ^4^He-OPMs sensors show deflection amplitudes of 248 fT and 270 fT for the radial and tangential components, respectively, a 3.8 to 4.1-fold increase. Further peaks occur around 200 and 300 ms (for SQUID-MEG 200 ms and 321 ms), which can only be reliably detected for the tangential ^4^He-OPMs axis (at 224 ms and 346 ms). Similarly, the cross-correlation of the RMS time courses reveals a correlation between SQUID-MEG and the radial component of ^4^He-OPMs of r = 0.80 and 0.71 for the tangential component.

Figure 7 directly compares the event-related fields for the SQUID and ^4^He-OPMs sensors for the visual stimulation task. For each subject, the SQUID and ^4^He-OPMs sensor (in both radial and tangential directions) with the highest signal-to-noise ratio was chosen. The top three panels show individual results with varying degrees of correspondence to the SQUID-MEG signal. Again, there is a high level of correspondence between sensor types, especially when noise levels are low, with correlations up to r = 0.86. For a complete overview of all subjects, and an estimate of the variability between subjects, see Figure A2 in the Appendix A.

SNR. As can be seen in Figure 8, SNR values again show a significant overlap between SQUID-MEG and ^4^He-OPMs. For the visual task, ^4^He-OPM SNR was a bit lower than in the somatosensory task (SNR 14.1 SD 4.9 for the radial axis), although SNR is nearly equal for both axes. SQUID-MEG benefits from the highest SNR (26.0 SD 8.4). This difference is significant (paired samples *t*-test: *t* = 5.8, *p* < 0.001). Note that also here, the range of SQUID-MEG SNR is larger than for the OPM sensors.

Oscillatory power. Figure 9 shows the time-frequency representation of SQUID-MEG and ^4^He-OPMs data for the visual task. Significant clusters, relative to baseline, are marked in black. Both SQUID-MEG and ^4^He-OPMs show a post-stimulus decrease in the alpha/beta (8–30 Hz) range and a simultaneously elevated gamma (>40 Hz) response, as expected with the presentation of visual Gabor stimuli. Although the gamma response was lower in the ^4^He-OPM sensors than in the SQUID sensors, there is significant power at the core gamma response frequency, suggesting a robust response, especially in the radial component.

## 4. Discussion

Here we present a direct comparison between conventional, cryogenic SQUID-MEG and newly developed wearable room-temperature ^4^He-OPM sensors using a somatosensory and visual stimulation paradigm. Results show encouraging similarities between the two modalities and significant improvements over previous results from the ^4^He-OPMs sensor.

Considerable gains have been made relative to the previous iteration of ^4^He-OPM sensors, described in [25]. The measured sensitivities of the ^4^He-OPM sensors of <43 fT/√Hz on two of the three axes are much better than the previously described 200 fT/√Hz. Event-related fields recorded using the ^4^He-OPMs sensors show a 3.8 to 4.5-fold increase in amplitude compared to SQUID-MEG, which is in line with the expected gain due to the reduced distance to the scalp for the ^4^He-OPM sensors and is equal or above those reported for alkali OPMs [4,5,6,7,39]. The current study used a flexible helmet to mount the ^4^He-OPM sensors. As the ^4^He-OPMs sensors work at room temperature, there is no limit to the proximity of the sensor to the scalp, and an optimized helmet design may increase signal amplitude even further.

Results from both tasks show a high degree of correlation between the time courses obtained using SQUID-MEG and ^4^He-OPMs. Deflections were highly similar in latency for both tasks, between the sensor types and present in both the radial and tangential components of the ^4^He-OPM signals. This is most pronounced in the tangential ^4^He-OPM component. Interestingly, the tangential ^4^He-OPM component also seemed to capture more late components of the visual response than the radial ^4^He-OPM component, suggesting that the tangential component may provide additional information as it may be sampled from different neuronal populations. These results are encouraging when taking into account the fact that we used a template subject to co-register the SQUID and OPM sensors leading to limited spatial accuracy.

Time-frequency decomposition of the visual stimulation data reveals significant alpha/beta and gamma oscillatory components that are highly similar across sensor types, showing that the ^4^He-OPM sensors can pick up changes in oscillatory brain dynamics across the human oscillatory neural range. However, while the absolute signal is higher in amplitude, the percent signal change from baseline is lower for ^4^He-OPM sensors compared to SQUID-MEG, especially in the higher oscillatory (gamma) range. Previous results have shown that gamma band activity can be reliably recorded using OPMs with relative power increases equal to or better than SQUID-MEG [10]. The reason for the lower performance in our study may not be due to a limitation of the sensor but rather due to the placement of the sensors on the posterior part of the head. Given that the cables connecting the OPM sensors were rather large and heavy in the current design, combined with free head movement, this may have caused tension in the neck area and subsequent muscle activity. Muscle contractions generally induce high-frequency (e.g., gamma) components in the data, which may have elevated the noise floor in this case.

There was a clear overlap in signal-to-noise ratios of conventional SQUID-MEG and ^4^He-OPMs, although SQUID-MEG yielded generally higher SNR. This is somewhat expected, as the ^4^He-OPMs sensor has an effective level of noise up to 45 fT/√Hz [26], or <43 fT/√Hz in our study, compared with a sensitivity of <3.4 fT/√Hz for SQUID-MEG. Of course, the decreased distance to the scalp is a mitigating factor, resulting in data quality nearing the fully developed SQUID-MEG. Besides intrinsic noise, there are important differences to consider when comparing performance between the ^4^He-OPM sensors and the CTF SQUID-MEG system used in this study: The OPM sensors were head-mounted, and any head movement produced by the participants caused movements of the sensors in a reduced, but non-homogeneous magnetic field. The (single) reference sensor, mounted 10 cm above the head, captures a significant portion of this movement-related signal, but as it may experience a slightly different field and its movements are slightly different from the scalp sensors, it is not as effective as the CTF integrated reference sensors in reducing environmental noise. Furthermore, the aforementioned cabling of the sensors, which will be optimized in the next iteration of the system, is likely to induce not only additional muscle contractions but also the slight movement of the sensors relative to the head. Although head movements were not recorded in the current study, we observed significant variation in data quality between participants, likely attributed to movement-related artifacts and/or sensor placement. Within the experimental group, some participants show ^4^He-OPM data quality equal to SQUID-MEG, suggesting that the ^4^He-OPM system is capable of higher performance than SNR values may currently reflect. Obviously, the ability to use head-mounted sensors with free head movement is a key benefit of using OPMs, but it may also explain the lower performance of the OPM system. An added benefit may be a more consistent distance to the scalp relative to SQUID-MEG, where there can be considerable variation in head placement and shape relative to the fixed sensor location [19]. This may explain the extended range of SNR values found for SQUID-MEG relative to the OPM system. Another factor to consider when comparing SQUID-MEG and OPM is the number of sensors used [40,41]. The current ^4^He-OPMs sensors featured four sensors on the scalp and one reference sensor. The SQUID-MEG recordings benefit from whole-head coverage with 275 sensors and an array of 29 reference sensors. This allows for many more degrees of freedom for artifact rejection compared to the limited means of artifact suppression for our ^4^He-OPM setup.

Note that in our study, we employed a simple regression-based technique to eliminate environmental noise and movement artifacts using the reference sensor. To further improve data quality, aside from increasing the number of sensors, more advanced techniques can be employed, such as homogenous field correction [42] and source reconstruction techniques, such as beamforming [43], to suppress signals of non-neural origin. By using a larger sample size than employed by previous studies, the current study was able to accurately estimate group-level correspondence between sensor types, as well as characterize sources of variability. The current results show a promising progression towards a high-quality, versatile MEG system without the drawbacks of conventional cryogenic SQUID-MEG. Future efforts will be focused on the implementation of a lightweight whole-head system, which will allow for greatly improved potential for noise reduction as well as extended head coverage.

## Figures and Tables

**Figure 1 sensors-23-02801-f001:**
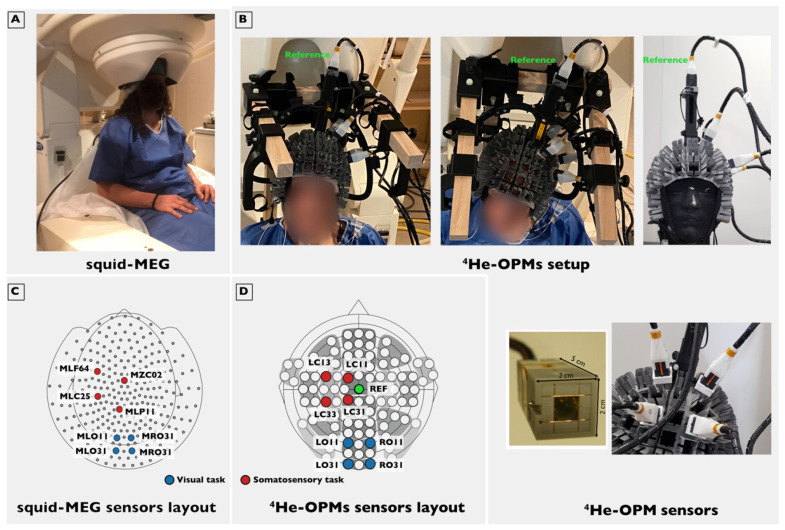
Experimental setup. (**A**) SQUID-MEG system used in this study with the subject in a typical seated position. (**B**): Top left: Subject setup with the five ^4^He-OPMs used in the somatosensory task. One of them serving as a reference sensor (green label) is placed over the top of the head, and the four other ones are located on the left side of the subject. The cables are supported by a wooden frame. Top right: Same setup on a phantom head without the wooden frame. Bottom right: zoomed view of one of the ^4^He-OPMs used and zoomed view of the sensors installed in the headset. The sensor has a 2 cm by 2 cm by 5 cm footprint. The glass cell containing the sensitive helium gas and the associated Helmholtz coils are visible. (**C**) SQUID-MEG sensors layout with the sensors closest to the OPMs location in red for the somatosensory task and in blue for the visual task. (**D**) ^4^He-OPMs sensors layout in red for the somatosensory task and in blue for the visual task.

**Figure 2 sensors-23-02801-f002:**
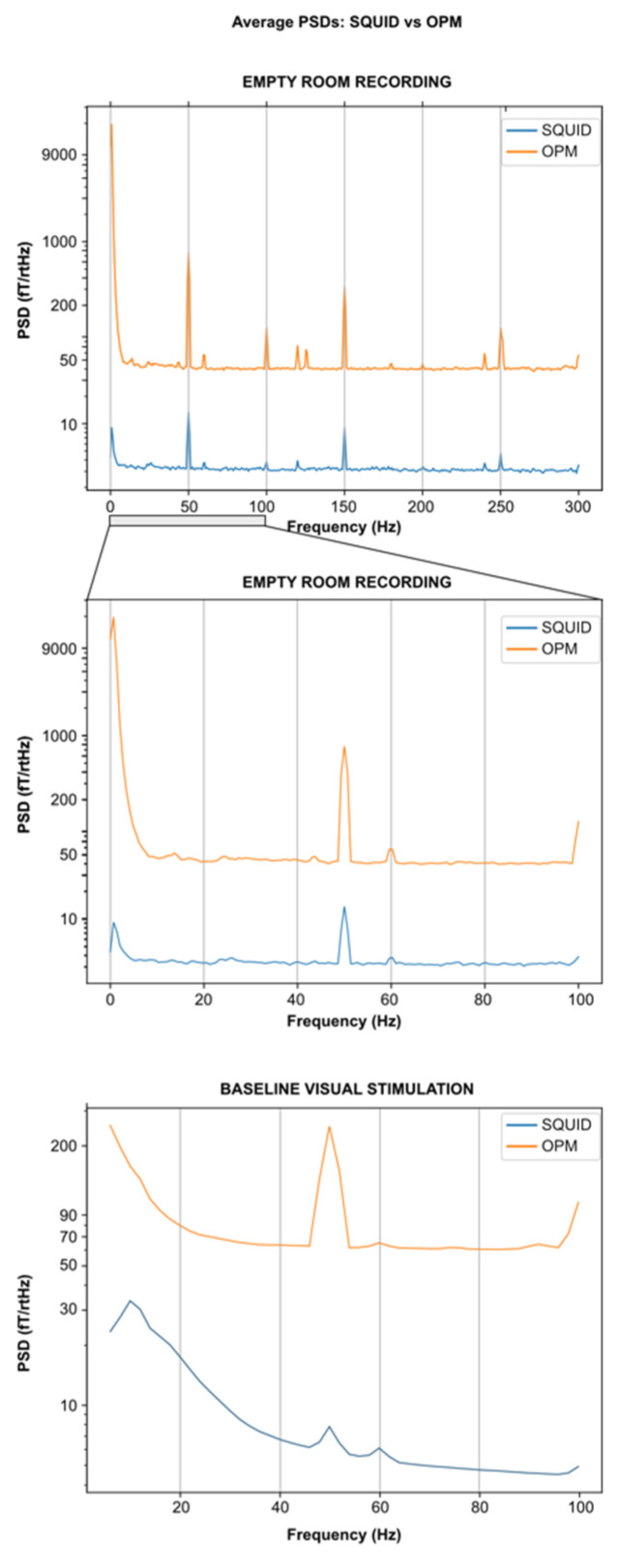
Empty room and visual task baseline average PSDs for SQUID-MEG and ^4^He-OPMs. Mean PSDs obtained after averaging PSDs over sessions and over all the sensors used in this study: SQUID-MEG: MLO31, MLO11, MRO21, MRO11, MLC11, MLC13, MLC33, MLC31 and ^4^He-OPMs: All 4 sensors except the reference with the two directions (radial and a tangential one) used in this study. No notch filters were applied for this figure. Top: Empty room full spectrum up to 300 Hz. Middle: Empty room spectrum zoomed up to 100 Hz. Bottom: Visual task baseline (500 ms) spectrum up to 100 Hz.

**Figure 3 sensors-23-02801-f003:**
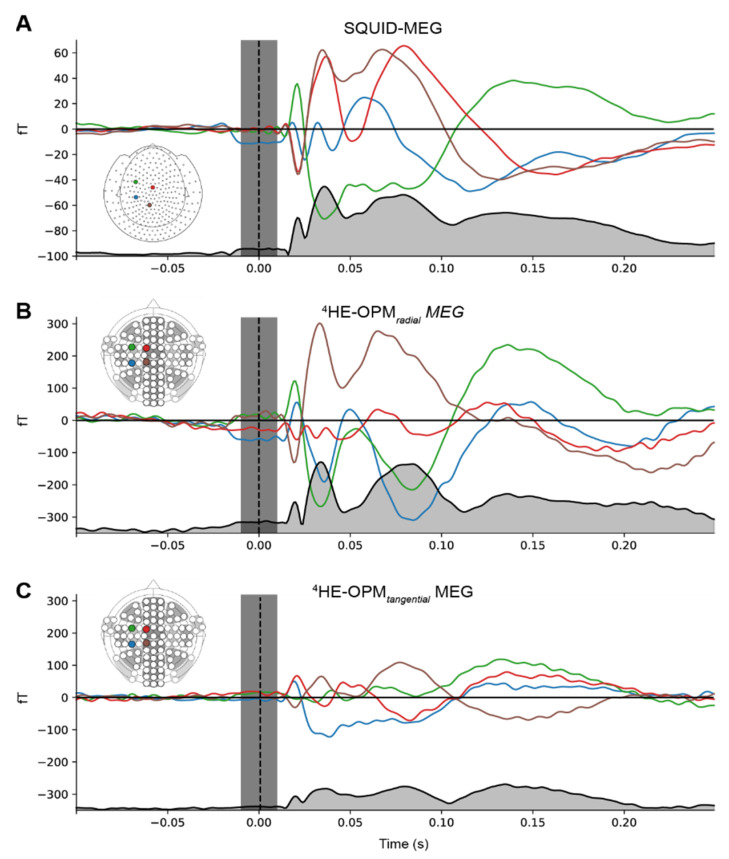
Event-related fields for SQUID-MEG (**A**), ^4^He-OPMs in the radial (**B**) and tangential axis (**C**). Gray-filled lines at the bottom of each panel represent the RMS of the combined signal. Gray vertical area denotes the suppressed stimulation artifact. Note that the scales for SQUID-MEG and ^4^He-OPMs are not the same.

**Figure 4 sensors-23-02801-f004:**
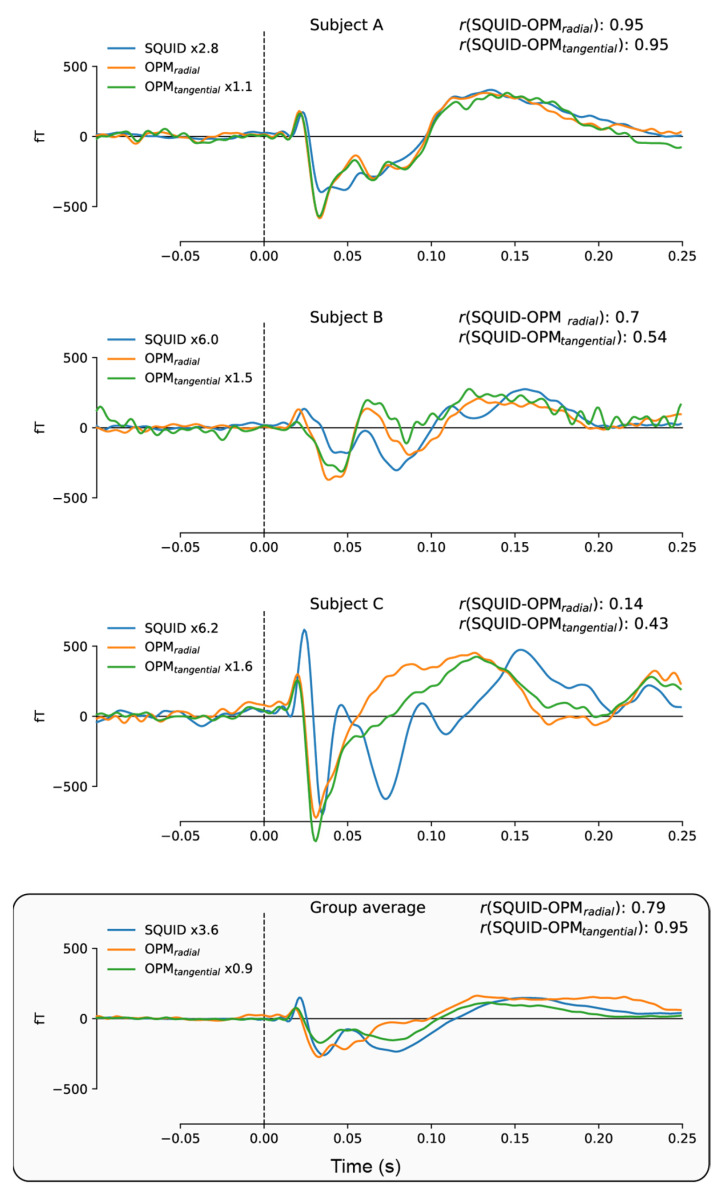
Individual time-courses of best SNR sensors following somatosensory stimulation for SQUID-MEG, radial ^4^He-OPMs and tangential ^4^He-OPMs. For visualization only, a multiplication factor and polarity alignment are applied to the SQUID-MEG and tangential axis of the ^4^He-OPMs sensors with reference to the radial axis ^4^He-OPMs. The top three panels depict three representative subjects with varying degrees of correlation between SQUID-MEG and ^4^He-OPMs. The bottom panel shows the group average (*n* = 17).

**Figure 5 sensors-23-02801-f005:**
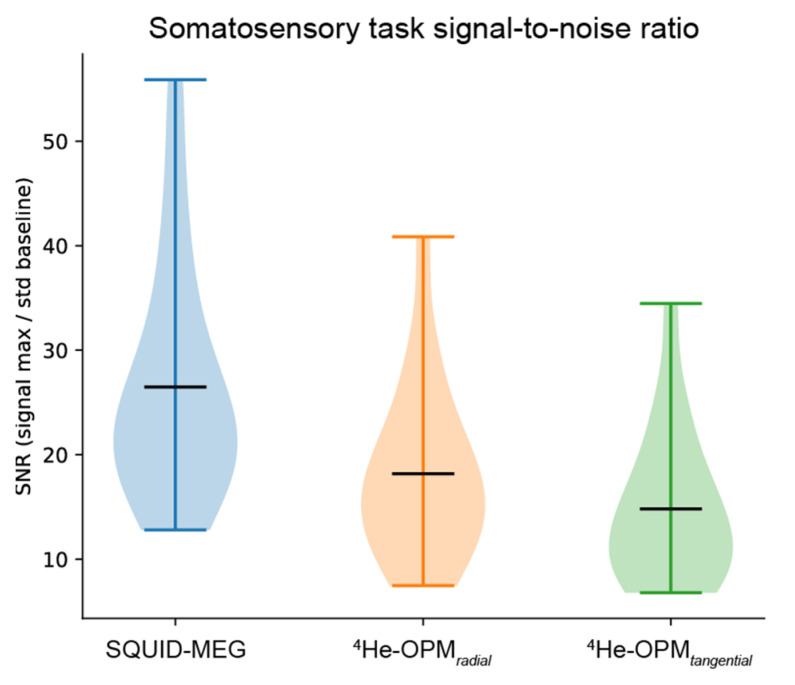
Average signal-to-noise ratio per modality, sensor type and axis. Black horizontal bars denote the group means. Plots span the entire data range.

**Figure 6 sensors-23-02801-f006:**
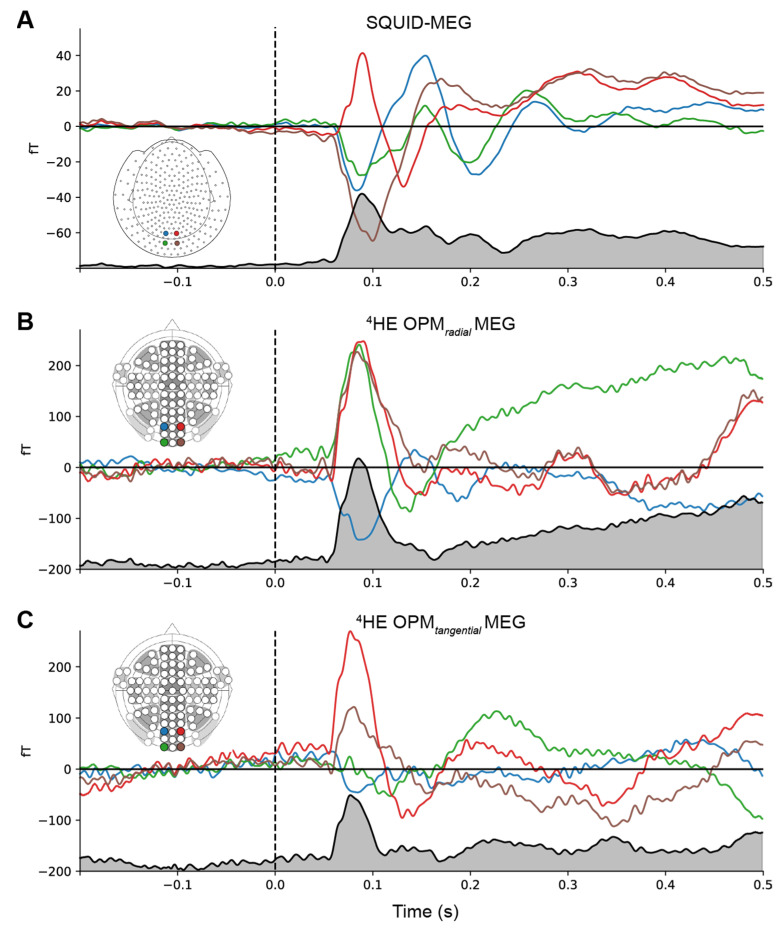
Group averaged event-related fields for conventional SQUID-MEG (**A**), ^4^He-OPMs in the radial (**B**) and tangential direction (**C**). Gray-filled lines at the bottom of each panel represent the RMS of the combined signal. Note that the scales for SQUID-MEG and ^4^He-OPMs are not the same.

**Figure 7 sensors-23-02801-f007:**
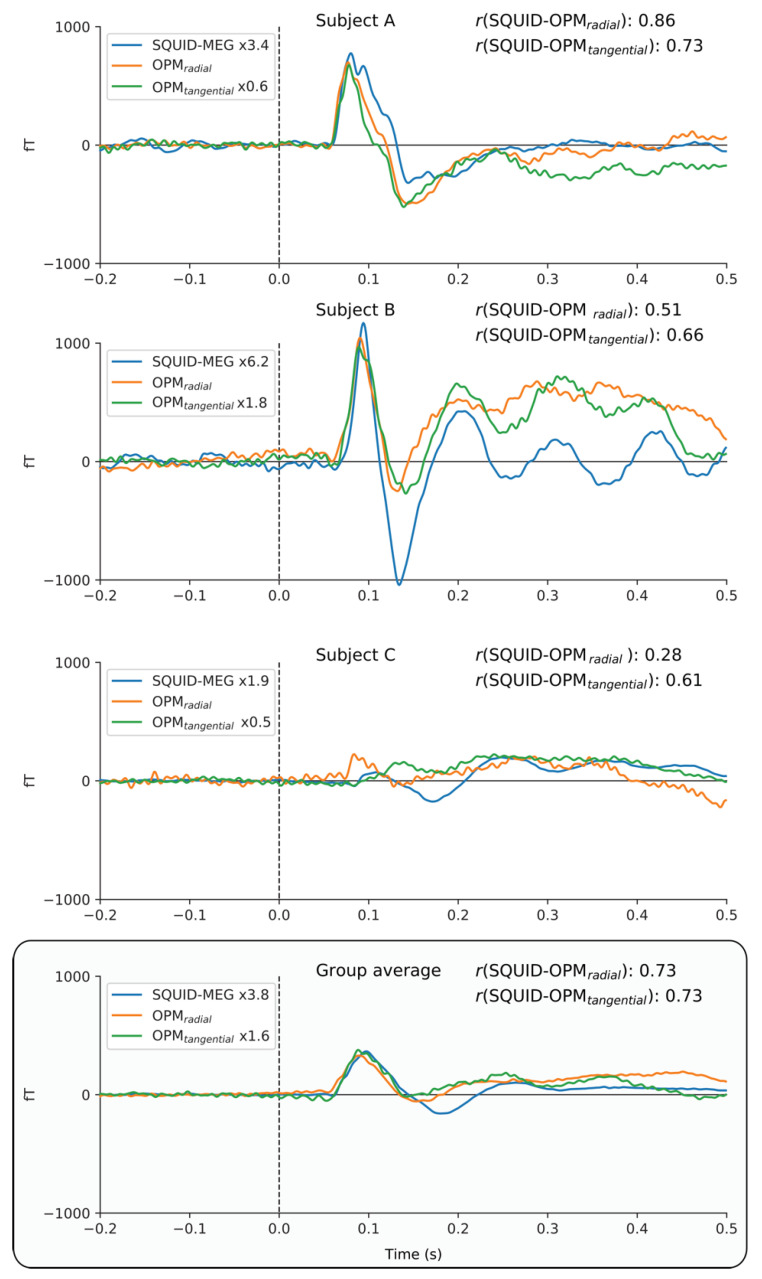
Individual time-courses of best SNR sensors following visual stimulation for SQUID, ^4^He-OPMs radial and ^4^He-OPMs tangential sensors. For visualization only, a multiplication factor and polarity alignment are applied to the SQUID-MEG and tangential ^4^He-OPMs with reference to the radial ^4^He-OPM sensor. The top three panels depict three representative subjects with varying degrees of correspondence between SQUID-MEG and ^4^He-OPMs. The bottom panel shows the group average (*n* = 18).

**Figure 8 sensors-23-02801-f008:**
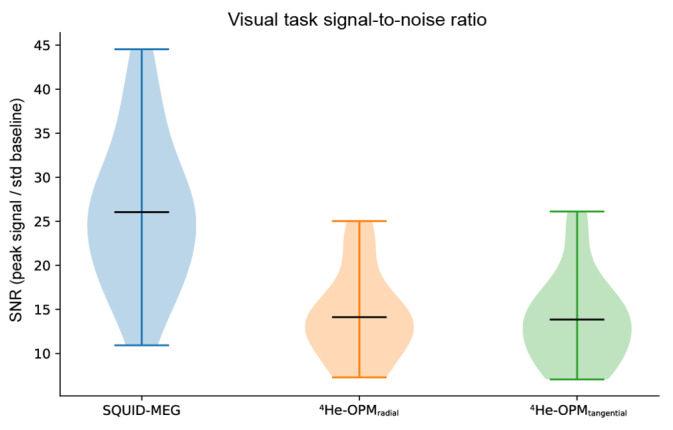
Average signal-to-noise ratio per modality, sensor type and axis, calculated as the maximum absolute post-stimulus onset deflection [0 s, 0.3 s] divided by the standard error of the baseline [−0.2 s, 0 s]. Black horizontal bars denote the group means. Plots span the entire data range.

**Figure 9 sensors-23-02801-f009:**
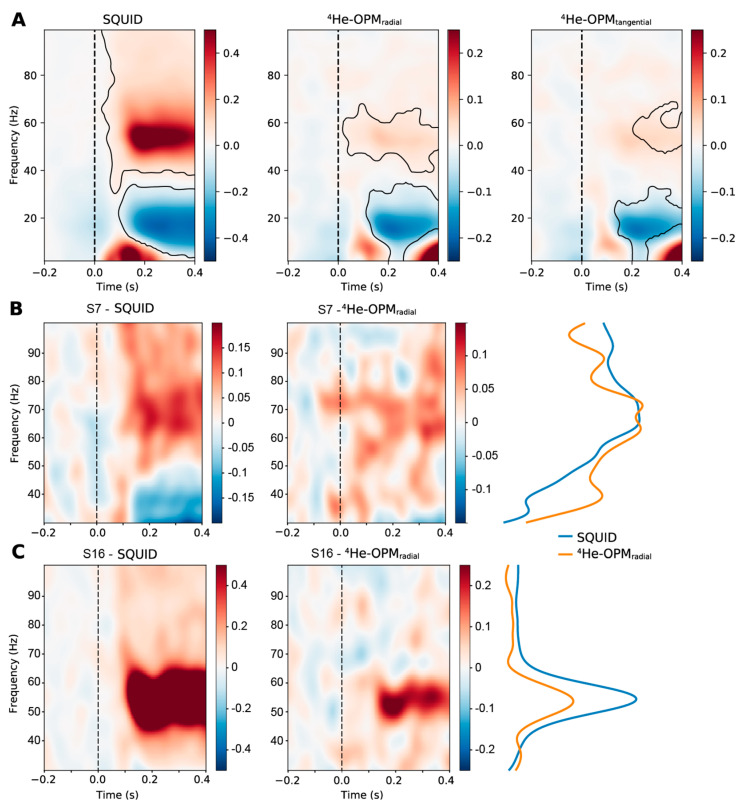
Group-average time-frequency representation of the visual experiment MEG data for the SQUID-MEG and ^4^He-OPM in the radial and tangential axes (**A**). Values denote the percent change relative to baseline [−0.4 s, 0 s]. Note that the scale is different between SQUID and ^4^He-OPMs sensors. Significant clusters (*p* < 0.05, two-tailed) are contained within areas marked in black. The onset of the visual stimulus was at t = 0. (**B**,**C**) depict time-frequency representations of two selected participants, one with a high individual gamma frequency (**B**) and a low to average frequency (**C**), in the gamma range for SQUID-MEG (**left**) and ^4^He-OPMs (radial axis, (**middle**)). Post-stimulus percent signal change [0.1 s, 0.4 s] is depicted on the (**right**) (scaling is adjusted for comparison).

## Data Availability

Anonymized data supporting the results of this study are available from the corresponding author upon reasonable request and validation by regulatory and ethical bodies and subject consent.

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
