# Peer review of "A New Generation of OPM for High Dynamic and Large Bandwidth MEG: The 4He OPMs—First Applications in Healthy Volunteers"

_sensors, 2023, doi:10.3390/s23052801_

Round 1

Reviewer 1 Report

This manuscripts describes a study comparing the performance of novel 4He-OPM sensors for MEG with conventional SQUID sensors in 18 subjects using somatosensory and visual evoked response paradigms. Similar event-related waveforms were observed with both types of sensors. The noise level of the OPM sensors was higher, but SNR values were comparable because the OPM sensors can be placed close to the brain sources and thus provide higher signal magnitude. The current 4He-OPM sensors had lower noise level than previously reported, and a major advantages over other types of OPM sensors is that they operate room temperature and therefore can be placed right against the scalp.

The results are promising for the development of non-cryogenic sensors for biomagnetism. At this stage of the sensor development, the present results indicate that the performance can be similar to that of the SQUIDs for studying brain activity.

I have the following relatively minor comments:

- line 13 (Abstract) and line 32 (Introduction): “MEG provides a direct measure of neuronal activity” This statement is somewhat questionable considering the known uncertainties related to source estimation. Better as “MEG provides a measure of electrical activity in the brain”

- line 24 (and 84): “will provide a detailed recording” Here, “detailed” is quite ambiguous, it would be better to say something more specific.

- line 51: ”allowing  a 3-8 fold increase in SNR” Please specify what is this increase is compared with. Should this say “signal” instead of “SNR”?

- line 113: “two of the three axes” IT would be of interest to make some comment about the noise level on the third axis, especially considering that one of the stated advantages of the sensors is the natively 3D vectorial measure

- line 140: “aged 18-60 (34.2+-8.2)” better as “aged 18-60 years (mean 34.2, std. [or s.e.m?] 8.2)”

- line 226: “empty room recording” It would be of interest to show the empty room spectrum, and also report how the empty room PSD compares to the pre-stimulus baseline levels in the event-related data. For the oscillatory activity, is the sensor noise larger than the background brain activity in the baseline?

- Figure 4: It would be helpful to use the same color code for the three types of sensors in Figures 3 and 4 (and 6 and 7).

- line 303: “SNR is in a comparable range” Calling  the SNRs “comparable” is somewhat subjective here, considering that the SQUID SNR was larger almost by a factor of 2.

- line 317; “the gamma response is attenuated” The wording here is potentially confusing, since the gamma was actually higher than the baseline level. Better as “the gamma response was lower in 4He-OPM than in SQUID-MEG”
